# The Relationship between the Presence of White Nails and Mortality among Rural, Older, Admitted Patients: A Prospective Cohort Study

**DOI:** 10.3390/healthcare9121611

**Published:** 2021-11-23

**Authors:** Ryuichi Ohta, Yoshinori Ryu, Chiaki Sano

**Affiliations:** 1Community Care, Unnan City Hospital, 699-1221 96-1 Iida, Daito-cho, Unnan 699-1221, Shimane Prefecture, Japan; yoshiyoshiryuryu.hpydys@gmail.com; 2Department of Community Medicine Management, Faculty of Medicine, Shimane University, 89-1 Enya cho, Izumo 693-8501, Shimane Prefecture, Japan; sanochi@med.shimane-u.ac.jp

**Keywords:** white nail, mortality, general medicine, older patient, rural, community hospital

## Abstract

White nails are a sign of various physical deteriorations, including poor nutrition, organ damage, and aging. During a physical examination, white nails can be a helpful health indicator in older patients with vague and multiple symptoms. In this prospective cohort study of patients admitted to the Department of General Medicine in a rural community hospital, we investigated the relationship between white nails and patient mortality. Patient data, including age, sex, condition, and bloodwork results, were collected. Trained family physicians confirmed the presence of white nails. Logistic regression analysis was performed to determine the relationship between white nails and death during hospitalization. Of 711 study participants, 74 died during hospitalization. White nails, male sex, and caregiver dependence were associated with high in-hospital mortality (odds ratio (OR) = 3.47, *p* < 0.001; OR 2.05, *p* = 0.01; OR 1.92, *p* = 0.049, respectively). High albumin concentration was associated with low in-hospital mortality (OR = 0.44, *p* < 0.001). White nails, along with serum albumin concentration, male sex, and caregiver dependence, are associated with mortality. The identification of white nails can predict the deterioration of patients. Various professionals should learn to identify the presence of white nails to facilitate the care of elderly patients.

## 1. Introduction

Changes in nail conditions can indicate various systematic changes in the human body, such as poor nutrition and chronic disease [1]. Nail color changes, which occur gradually, can occur in nail plates or nail beds [2,3]. These changes can indicate chronic changes in the human body [4,5,6]. Patients with heart failure, liver cirrhosis, and advanced age experience whitening of the nails, with the distal parts of the nails turning brown due to telangiectasis; this condition is called Terry’s nail [7,8]. A poor nutritional status can progressively cause Terry’s nail [9]. On the other hand, whitening of the nails with brown distal nail parts due to melanin deposition is called Lindsay’s nail; this is indicative of malnutrition and renal disease [10]. The mechanism underlying Terry’s and Lindsay’s nails is not fully understood. Given that nail plates and nail bed conditions may not change quickly in response to acute diseases, these changes can be called static physical findings [11]. The state of patients’ nails may be indicative of their nutritional and chronic disease status. Given that nutritional conditions can affect the clinical progression of patients with acute diseases, nail health can be used as a predictive risk factor for patients’ morbidity and mortality.

The examination of nails has been considered helpful for clinical reasoning in clinical settings. However, in examinations of older patients, differentiation of the findings of nail examination can be difficult. One of the reasons for this is multimorbidity. Older patients often have multiple chronic diseases that affect their health conditions, which modify their physical presentation and examination findings [12]. Specific nail examination findings can indicate several possible medical conditions and cannot reveal the presence of a particular disease [2]. Aging can change nail plate and nail bed conditions due to atherosclerosis and fibrosis of the peripheral tissues [13]. In these circumstances, including Lindsay’s or Terry’s nails, nail color typically changes to white. Both conditions reportedly indicate the presence of liver and renal diseases [7,10]. However, differentiating between Lindsay’s and Terry’s nails is difficult because these names are used interchangeably in previous research [7,10]. Furthermore, although aging can affect nail conditions, the prevalence of these changes in the older generation is not well understood [10]. In countries with older populations, nail examination should be considered and investigated for use in clinical situations. Abnormal nail findings should be defined (i.e., a white nail without a lunula) [11]. Currently, there is no information on the distribution of nail changes or whether the changes are transient. Nail changes are more common in the nails of all fingers except the nail of the first finger. Changes can be present mainly in the nail beds of fingers associated with various chronic diseases [11]. Thus, during the physical examination of patients with systemic conditions, physicians can note nail changes on all digits.

The presence of white nails can provide critical information in older patients and may be related to their prognoses. White nails can be the result of various biomedical conditions, such as renal and liver disease or a poor nutritional condition, which can be related to morbidity and mortality in patients [11]. Therefore, white nails can be related to patients’ clinical outcomes. However, no studies have investigated the specific relationship between the presence of white nails and patient health outcomes. Nail conditions can change with aging, and the prevalence of white nails can change among the elderly [14]. Acute changes in body fluids, dynamic physical findings, and laboratory data can make the prediction of general conditions difficult. The presence of white nails may indicate the condition and vulnerability of older patients at admission and is not affected by acute changes in medical conditions. However, there are no studies regarding the relationship between white nails and readmission and mortality rates. Therefore, we sought to determine the relationship between the presence of white nails and in-hospital mortality among older patients. As societies age, comprehensive assessments of older patients’ conditions, including an examination of nail conditions, can be vital for rural and home care due to the lack of healthcare resources. The purpose of this study was to investigate the relationship between white nails and mortality rates among older patients in rural community hospitals.

## 2. Materials and Methods

This study was a prospective cohort study of patients admitted to the Department of General Medicine in a rural community hospital.

### 2.1. Setting

Unnan City is one of the most rural cities in Japan and is located in the southeast of Shimane Prefecture. In 2020, the total population of Unnan was 37,638 (18,145 males and 19,492 females), with 39% aged over 65 years, which is expected to reach 50% by 2025. There are 16 clinics, 12 home care stations, three visiting nurse stations, and only one public hospital (Unnan City Hospital) [15]. At the time of this study, Unnan City Hospital had 281 beds comprising 160 acute care beds, 43 comprehensive care beds, 30 rehabilitation beds, and 48 chronic care beds. There were 14 medical specialties, and the nurse-to-patient ratio was 1:10 for acute care, 1:13 for comprehensive care, 1:15 for rehabilitation, and 1:25 for chronic care. Patient care was managed by family physicians and nurses in the acute care division. Patients were regularly followed up at Unnan City Hospital or other medical institutions in Unnan City from 1 April 2020 to 30 June 2021. All readmissions were performed at the Unnan City Hospital [16].

### 2.2. Participants

All patients over 65 years of age who were admitted to Unnan City Hospital and discharged from or died at the hospital from 1 April 2020 to 31 March 2021 were included in this study. The study period was from 1 April 2020 to 30 June 2021 to observe the mortality rate. Patients who lost their fingers, had fungal infections of the nails, had traumatic nail deformations, had white spots warranting referral to the dermatologist, underwent potassium hydroxide testing, and those referred to other specialties were excluded from the present study.

### 2.3. Measurements

#### 2.3.1. Assessment of White Nails

White nails were characterized by a white appearance of the proximal nails without a lunula [11]. Changes in the distal part of the nail caused by deposition of melanin were not considered in the assessment of white nails [7,10]. The nail lunula can be diminished in all fingers, but the lunula of the first finger is often persistent, even in older individuals. All assessors evaluated patients’ nails for the appearance of whitening of the nails upon admission and were unaware of the endpoints of the patients. To improve the precision of the diagnosis of white nails, the family physicians who performed the nail examinations in this study were trained and tested using clinical pictures of white nails from a previous study [11]. During training, family physicians identified and discussed white nails while examining patients using the previous study as a reference [11]. This training period lasted approximately three months before patients were enrolled into the study [11]. During the first month of training, the family physicians read previous articles to learn about while nails and observed their patients. In regular clinical rounds, attending family physicians presented the conditions of the patients’ nails and discussed the presence of apparent whitening, using the definitions in this study as references. This process continued for the following two months to improve their ability to detect the presence of white nails. To improve the reliability of the diagnoses, participants were evaluated by more than two family physicians. If there were discrepancies between the assessors, additional assessors examined the participants and discussed the appearance of their nails until an agreement was reached.

#### 2.3.2. The Patients’ Data

The participants’ background information was extracted from the electronic health records of Unnan City Hospital. Data were collected concerning the participants’ age, sex, body mass index (BMI), albumin concentration, serum creatinine levels, estimated glomerular filtration rate (eGFR), and care level based on the Japanese long-term insurance system [17]. In addition, Charlson comorbidity indices (CCIs) based on past medical histories (the presence of heart failure, myocardial infarction, asthma, chronic obstructive pulmonary diseases, kidney diseases, liver diseases, diabetes mellitus, brain infarction, brain hemorrhage, hemiplegia, connective tissue diseases, dementia, and cancer) [18], admission duration, and death during admission were assessed.

#### 2.3.3. Analysis

The student’s t-test was performed on parametric data, and the Mann–Whitney U test was performed on non-parametric data. Based on previous studies and the average of variables, numerical variables were dichotomized as follows: CCI (≥5 and <5) [18] and care level (≥1 and 0) based on the burden on caregivers and families [17]. A univariate regression model was used to assess whether mortality was associated with the independent variables, including the presence of white nails. Variables with statistically significant differences in the univariate regression analysis were further analyzed using a logistic regression model. Regarding the sample size calculation, 219 participants would be needed with 80% statistical power and 5% type 1 error to detect a difference in the percentage of death of 10% between the white nail and non-white nail groups. Cases with missing data were excluded from the analysis. Statistical significance was defined as a *p*-value < 0.05. All statistical analyses were performed using EZR (Saitama Medical Center, Jichi Medical University, Saitama, Japan), which is a graphical user interface for R (The R Foundation, Vienna, Austria).

#### 2.3.4. Ethical Considerations

The hospital was assured of the anonymity and confidentiality of the patients’ information. Information about the study was posted on the hospital website without disclosure of the patients’ details. To address any questions regarding this study, the contact information of the hospital representative was also posted on the website. All participants were informed about the purpose of this study, and written informed consent was obtained from all participants or their families. The Clinical Ethics Committee of our institution approved this study (Approved date: 1 March 2020).

## 3. Results

### 3.1. The Demographic of the Participants

Of the 2894 patients admitted to the community hospital, 748 were admitted to the Department of General Medicine. After excluding 37 patients with missing data, 711 participants were evaluated. The patient inclusion flowchart is presented in Figure 1. Of the participants, 29.8% had white nails. The average participant age was 81.95 (standard deviation (SD) = 13.63). Of the patients evaluated, 44.4% were male. Statistically significant differences in age, sex, albumin concentration, BMI, the presence of white nails, CCI, caregiver dependence, and admission duration were present when comparing patients who died during the study period with those who did not (Table 1).

### 3.2. The Relationship between the Presence of White Nails and Mortality

A multivariate logistic regression model was used to determine the association between the independent variables and in-hospital mortality. The independent variables included age, sex, albumin concentration, BMI, presence of white nails, CCI, caregiver dependence, and admission duration. The presence of white nails, (odds ratio (OR) = 3.47, *p* < 0.001), male sex (OR = 2.05, *p* = 0.01), and caregiver dependance (OR = 1.92, *p* = 0.049) were associated with high in-hospital mortality. Conversely, a high albumin concentration was associated with low in-hospital mortality (OR = 0.44, *p* < 0.001). There was no statistically significant relationship between the other variables assessed and in-hospital mortality (Table 2).

### 3.3. Reasons for Hospital Admission

The most common reason for hospital admission was heart failure (11.8%), followed by urinary tract infection (11.5%), brain stroke (8.2%), bacterial pneumonia (6.0%), and aspiration pneumonia (5.9%) (Table 3).

Other reasons for admission included tonsillitis, sudden deafness, superior mesenteric artery (SMA) syndrome, spontaneous bacterial peritonitis, sarcoidosis, renal failure, pulmonary embolism, paroxysmal supraventricular tachycardia, polymyositis, pleuritis, parotitis, non-tuberculosis mycobacterium, microscopic polyangiitis, liver cirrhosis, infectious mononucleosis, hyperthermia, human immunodeficiency virus (HIV), hepatitis, hemoptysis, Hashimoto encephalitis, fatigue, functional dyspepsia, erysipelas, eosinophilic gastritis, dyspnea, drowning, depression, chronic pleuritis, chronic fatigue syndrome, bronchitis, atrial fibrillation, asthma, amyotrophic lateral sclerosis, adrenal insufficiency, and acute disseminated encephalomyelitis.

## 4. Discussion

This prospective cohort study demonstrated that the presence of white nails was associated with mortality during hospital admission among older rural patients. The effect remained after adjusting for other independent variables related to mortality in hospitals. Among older people, serum albumin concentration, male sex, and caregiver dependence were found to be related to in-hospital mortality. The reasons for admission were common diseases among older patients. This study indicates that the consideration of nail condition during the physical examination of older people is an informative diagnostic assessment tool.

The importance of examining nails for whitening should be emphasized when examining older hospitalized patients to predict their risk of mortality. This study shows the relationship between white nails and mortality rates among older admitted patients. In general, older patients present with various symptoms and physical findings related to their medical histories and nutritional status [19,20]. Therefore, clinicians should consider the conditions of older patients comprehensively, including information on symptoms, physical findings, and other clinical data in their assessment [19,21]. Compared to other physical findings of major organs, such as the lungs and heart, nails and hands can be quickly examined in outpatient clinics and during bedside examinations without the need for sophisticated equipment or diagnostic tools. In addition, mortality in older patients can be affected by nutrition, age, and previous medical conditions [22,23]. As this study shows, the presence of white nails is an independent factor related to patient mortality and is most likely interrelated to these factors. Therefore, clinicians must be aware of nail conditions and their relationship to the worsening of more serious conditions. The pathophysiology of white nails can be explained by various factors other than nutrition as changes in the color of the nail plate can be the result of vascular changes or substance deposition [1,14,24]. In this study, the presence of liver and renal diseases was not related to the mortality rate in hospitals. The presence of white nails, which can be transient, may not be due to liver and kidney diseases [11]. Future studies should be carried out to elucidate the pathophysiology of white nails in depth. Furthermore, if vascular conditions can cause nail whitening, the color of the nail can change by increasing peripheral perfusion, which often alleviates the symptoms and conditions of diseases. Future studies should investigate the relationship between restorative changes in nail color upon admission and mortality among older patients.

Various clinical factors are related to the mortality of older patients admitted to community hospitals. First, serum albumin levels may be strongly related to the mortality of older admitted patients. Serum albumin is affected by various conditions, such as infections, body volume conditions, and nutritional conditions [25,26]. Low serum albumin concentrations can indicate poor nutritional conditions, which can lead to the inability to make albumin. Severe infections can lead to an increase in immunoglobulin that, in turn, suppresses albumin production, both of which can cause high mortality [27]. Second, males are more likely to die comparatively. Statistically, the longevity of men is lower in developed countries, including Japan [28,29]. Compared to women of the same age, men have higher mortality rates upon admission for acute diseases [29]. Furthermore, men tend to endure their symptoms until they become severe. This hesitancy to seek medical attention or help-seeking behavior (HSB) is related to a poor clinical course and high mortality [30]. Finally, caregiver dependence can be associated with higher mortality rates. Patients with caregiver dependence must receive help from others to receive medical care, so their HSBs are dependent on their caregivers and healthcare professionals [31]. According to previous studies, caregivers and home care professionals have difficulty judging the severity of older patients’ symptoms [32,33,34,35]. These challenges can cause a delay in seeking medical care, increasing the severity of conditions in caregiver dependent patients.

Common reasons for admission reflect present medical conditions in developed countries. In this study, the most common disease was heart failure (11.8%), followed by urinary tract infection (11.5%), brain stroke (8.2%), bacterial pneumonia (6.0%), and aspiration pneumonia (5.9%), which are common among older people [36,37,38]. Aging societies have changed the prevalence of diseases in communities, and the need for community hospitals is expanding. Organ-specific specialists cannot manage the increase in the number of heart failures and other cardiovascular diseases. General physicians must manage various diseases in community hospitals, especially in rural settings where fewer specialists are available. Previous studies have shown that the implementation of general medical facilities in rural communities can improve comprehensive care with improved quality of life in rural areas, meaning that general physicians can accommodate various diseases to some extent [39,40,41]. To support older peoples’ lives in homes and hospitals, an adequate understanding of each patient’s condition is vital. Thus, examination for the presence of white nails can be helpful for general physicians and various healthcare professionals in assessing the overall health of the individual. Studies should be performed to investigate the relationship between the presence of white nails and hospital admission or readmission rates.

One limitation of this study is the study setting. This study was carried out in a single, rural community hospital that primarily cares for older patients. Other hospital locations may have younger patients to whom these research results cannot be applied. However, considering the current aging of societies in rural areas, the results of this study can be used in various contexts in rural settings. In addition, we diligently followed all of the admitted patients, achieving a low rate of missing data, which would contribute to our findings’ reliability. Another limitation is the reliability of the diagnosis of white nails. In this study, family physicians who were trained to identify white nails performed all diagnoses. We focused only on the disappearance of the lunula and whitening of the proximal part of the nail. As there are variations in the whitening of nails and other types of nails, future studies can investigate the extent of whitening and other nail conditions by using transfer learning methods, and the mortality and morbidity of patients in these contexts [42]. Furthermore, the results of the study may be limited by the lack of blinding to the medical histories and the possibility of potential confounding factors affecting the assessment of the white nails. Future studies can evaluate the impact of knowing the patient’s history on white nail diagnosis. Potential confounding factors, such as the patient’s lifestyle history and the environmental conditions that impact nails, also need to be studied.

## 5. Conclusions

This study indicates that white nails are associated with mortality among older patients admitted in rural settings. Among this group, serum albumin concentration, male sex, and caregiver dependence are also related to mortality. Recognizing the presence of white nails can contribute to the awareness of the severity of individual cases. The identification of white nails can predict the deterioration of elderly patients’ conditions. In addition, various professionals can learn to identify the presence of white nails to facilitate the care of older patients.

## Figures and Tables

**Figure 1 healthcare-09-01611-f001:**
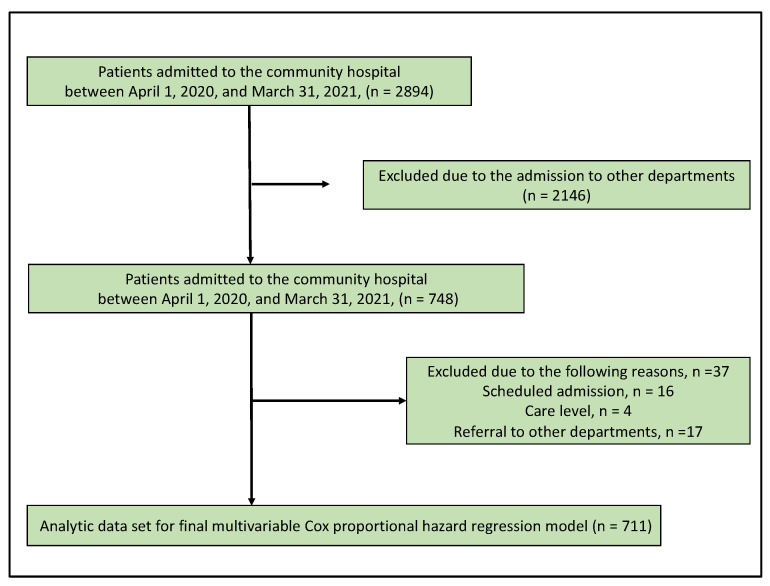
The flow chart of patient selection.

**Table 1 healthcare-09-01611-t001:** Patient demographics.

		Death		
Factor	Total	Yes	No	*p*-Value
n	711	74	637	
Age (years), mean (SD)	81.95 (13.63)	88.41 (6.59)	81.20 (14.03)	<0.001
male sex (%)	316 (44.4)	41 (55.4)	275 (43.2)	0.049
albumin, mean (SD)	3.49 (0.66)	2.96 (0.54)	3.55 (0.64)	<0.001
height, mean (SD)	153.37 (10.44)	152.73 (9.91)	153.45 (10.51)	0.577
body weight, mean (SD)	48.69 (11.69)	44.51 (8.67)	49.18 (11.90)	0.001
BMI, mean (SD)	20.59 (3.93)	19.10 (3.34)	20.76 (3.96)	0.001
creatinine	1.05 (1.09)	1.07 (0.78)	1.04 (1.12)	0.865
eGFR	58.75 (22.41)	57.85 (25.78)	58.86 (22.01)	0.714
white nail (%)	212 (29.8)	54 (73.0)	158 (24.8)	<0.001
Admission duration (IQR])	15.00 (1.00, 167.00)	24.50 (1.00, 146.00)	15.00 (1.00, 167.00)	<0.001
CCI ≥ 5 (%)	423 (59.5)	55 (74.3)	368 (57.8)	0.006
CCI (%)				
0	26 (3.7)	0 (0.0)	26 (4.1)	
1	17 (2.4)	0 (0.0)	17 (2.7)	
2	28 (3.9)	0 (0.0)	28 (4.4)	
3	50 (7.0)	2 (2.7)	48 (7.5)	
4	167 (23.5)	17 (23.0)	150 (23.5)	
5	147 (20.7)	15 (20.3)	132 (20.7)	
6	112 (15.8)	15 (20.3)	97 (15.2)	
7	92 (12.9)	12 (16.2)	80 (12.6)	
8	34 (4.8)	4 (5.4)	30 (4.7)	
9	21 (3.0)	2 (2.7)	19 (3.0)	
10	7 (1.0)	3 (4.1)	4 (0.6)	
11	5 (0.7)	3 (4.1)	2 (0.3)	
12	2 (0.3)	0 (0.0)	2 (0.3)	
13	1 (0.1)	0 (0.0)	1 (0.2)	
14	1 (0.1)	1 (1.4)	0 (0.0)	
15	1 (0.1)	0 (0.0)	1 (0.2)	
heart failure (%)	126 (17.7)	17 (23.0)	109 (17.1)	0.202
MI (%)	56 (7.9)	9 (12.2)	47 (7.4)	0.168
asthma (%)	36 (5.1)	4 (5.4)	32 (5.0)	0.782
peptic ulcer (%)	60 (8.5)	2 (2.7)	58 (9.1)	0.074
kidney disease (%)	55 (7.7)	5 (6.8)	50 (7.8)	1
liver disease (%)	26 (3.7)	2 (2.7)	24 (3.8)	1
COPD (%)	41 (5.8)	8 (10.8)	33 (5.2)	0.062
DM (%)	106 (14.9)	9 (12.2)	97 (15.2)	0.605
brain infarction (%)	128 (18.0)	11 (14.9)	117 (18.4)	0.525
brain hemorrhage (%)	53 (7.5)	7 (9.5)	46 (7.2)	0.482
hemiplegia (%)	23 (3.2)	2 (2.7)	21 (3.3)	1
connective tissue disease (%)	26 (3.7)	2 (2.7)	24 (3.8)	1
dementia (%)	131 (18.4)	16 (21.6)	115 (18.1)	0.432
cancer (%)	141 (19.8)	27 (36.6)	114 (17.9)	0.001
caregiver dependence (%)	313 (44.0)	55 (74.3)	258 (40.5)	<0.001
care level (%)				
0	398 (56.0)	19 (25.7)	379 (59.5)	
1	51 (7.2)	4 (5.4)	47 (7.4)	
2	83 (11.7)	16 (21.6)	67 (10.5)	
3	78 (11.0)	15 (20.3)	63 (9.9)	
4	57 (8.0)	11 (14.9)	46 (7.2)	
5	44 (6.2)	9 (12.2)	35 (5.5)	

Abbreviations: BMI, body mass index; eGFR, estimated glomerular filtration rate; CCI, Charlson comorbidity indices; MI, myocardial infarction; COPD, chronic obstructive pulmonary diseases; DM, diabetes mellitus; IQR, interquartile range; SD, standard deviation.

**Table 2 healthcare-09-01611-t002:** Multivariate logistic regression model for the relationship between in-hospital mortality and independent variables.

Factor	Odds Ratio	95% CI	*p*-Value
Age	1.03	1.00–1.07	0.069
Male sex	2.05	1.18–3.53	0.01
Albumin concentration	0.44	0.27–0.72	<0.001
BMI	0.97	0.90–1.04	0.39
CCI ≥ 5	0.92	0.49–1.72	0.79
Caregiver dependence	1.92	1.00–3.67	0.049
White nail	3.47	1.85–6.49	<0.001
Admission duration	1.00	1.00–1.01	0.5

Abbreviations: BMI, body mass index; CCI, Charlson comorbidity index.

**Table 3 healthcare-09-01611-t003:** The prevalence of the reasons for admission.

Disease	N	Percentage	Disease	N	Percentage
heart failure	84	11.8%	Hypothyroidism	5	0.7%
urinary tract infection	82	11.5%	Hypoglycemia	5	0.7%
brain stroke	58	8.2%	hepatic encephalopathy	5	0.7%
bacterial pneumonia	43	6.0%	*Clostridium difficile* colitis	5	0.7%
aspiration pneumonia	42	5.9%	vitamin B1 sufficiency	5	0.7%
cancer	23	3.2%	transient ischemic attack	4	0.6%
brain hemorrhage	20	2.8%	myocardial infarction	4	0.6%
trauma	19	2.7%	Dehydration	4	0.6%
syncope	18	2.5%	gastroesophageal reflux disease	4	0.6%
pseudogout	17	2.4%	varicella zoster virus infection	3	0.4%
sepsis	17	2.4%	Type 2 respiratory failure	3	0.4%
gastrointestinal bleeding	16	2.3%	temporal arteritis	3	0.4%
epilepsy	15	2.1%	septic vertebritis	3	0.4%
cellulitis	14	2.0%	Pancreatitis	3	0.4%
chronic obstructive lung disease	11	1.5%	diabetic ketoacidosis	3	0.4%
ischemic colitis	11	1.5%	polymyalgia rheumatica	3	0.4%
peripheral vertigo	11	1.5%	urinary stone	2	0.3%
fever	9	1.3%	Tuberculosis	2	0.3%
unconsciousness	9	1.3%	Tsutsugamushi	2	0.3%
loss of appetite	8	1.1%	normal pressure hydrocephalus	2	0.3%
acute enteritis	8	1.1%	liver abscess	2	0.3%
cholangitis	7	1.0%	iron deficiency anemia	2	0.3%
Parkinson’s syndrome	6	0.8%	Headache	2	0.3%
angina	6	0.8%	Guillain–Barré syndrome	2	0.3%
peptic ulcer	6	0.8%	Constipation	2	0.3%
medication-induced	6	0.8%	Anaphylaxis	2	0.3%
electrolyte disturbance	6	0.8%	Alcoholism	2	0.3%
pneumothorax	5	0.7%	anterior cutaneous nerve entrapment syndrome	2	0.3%
Ileus	5	0.7%	hypertensive emergency	2	0.3%
meningitis	5	0.7%	Others	36	5.1%

## Data Availability

The datasets used and/or analyzed during the current study may be obtained from the corresponding author upon reasonable request.

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
