# Peer review of "The Relationship between the Presence of White Nails and Mortality among Rural, Older, Admitted Patients: A Prospective Cohort Study"

_healthcare, 2021, doi:10.3390/healthcare9121611_

Round 1

Reviewer 1 Report

Your statistical data does not seem to be very clearly stated. There is no data showing the actual statistical difference. Other than the demographics of your cohort. 

Please explain why you had to publish patients' (deidentified) data on hospital website. 

There is lots of redundant material in the introduction, some of which is wrong, as stated to the authors in my comments. 

English language editing is required. 

Terry and Lindsay nails are clearly distinguished conditions. 

Too many repetitions on white nails in your text. 

Author Response

Responses to the Reviewers’ comments

Thank you very much for reviewing our manuscript and providing suggestions to improve on our manuscript. We have provided the point-by-point responses to the Reviewers’ comments; our revisions are in red font. Please consider our manuscript for publication.

Reviewer 1

Your statistical data does not seem to be very clearly stated. There is no data showing the actual statistical difference. Other than the demographics of your cohort. 

Response:

Thank you for this insightful comment. We agree with you, and we have revised the results by adding the statistical significance data for the variables in the multivariate logistic regression model in Tables 1 and 2, and not only the demographics data. Furthermore, we provided the following information in the results.

Line181 to 189

The multivariate logistic regression model was used to determine the association between the independent variables and in-hospital mortality. The independent variables included age, sex, albumin concentration, BMI, presence of white nails, CCI, and caregiver de-pendence. The presence of white nails was associated with high in-hospital mortality (odds ratio [OR] = 3.51, p <0.001). Male sex was associated with high in-hospital mortality (OR = 2.02, p = 0.011). High albumin concentration was associated with low in-hospital mortality (OR = 0.44, p <0.001). Caregiver dependence was associated with high in-hospital mortality (OR = 1.92, p = 0.049). There was no statistically significant relationship between the other variables assessed and in-hospital mortality.

Please explain why you had to publish patients' (deidentified) data on hospital website. 

Response:

Thank you for this question. We have revised the ethical considerations section as follows:

Line 156 to 162

The hospital was assured of the anonymity and confidentiality of the patients’ information. Information about the study was posted on the hospital website without disclosure of the patients’ details. To address any questions regarding this study, the contact information of the hospital representative was also posted on the website. All participants were informed about the purpose of this study, and written informed consent was obtained from all participants or their families. The Clinical Ethics Committee of our institution approved this study (Approved date: March 1, 2020).

There is lots of redundant material in the introduction, some of which is wrong, as stated to the authors in my comments. 

Response:

Thank you for this remark. We agree with you, and we have revised the introduction to improve its accuracy and length.

Line 33 to 45

Patients with heart failure, liver cirrhosis, and advanced age experience whitening of the nails, with the distal parts of the nails turning brown, due to telengiectasis; this condition is called Terry's nail [7, 8]. A poor nutritional status can progressively cause Terry’s nail [9]. On the other hand, whitening of the nails with brown distal nail parts due to melanin deposition is called Lindsay's nail; this is indicative of malnutrition and renal disease [10]. The mechanism underlying Terry's and Lindsay’s nails is not fully understood. Given that nail plates and nail bed conditions may not change quickly in response to acute diseases, these changes can be called static physical findings [11]. The state of patients’ nails may be indicative of their nutritional and chronic disease status. Given that nutritional conditions can affect the clinical progression of patients with acute diseases, nail health can be used as a predictive risk factor for patients' morbidity and mortality.

Line 52 to 66

Aging can change nail plate and nail bed conditions, due to atherosclerosis and fibrosis of the peripheral tissues [13]. In these circumstances, including Lindsay's or Terry's nails, nail color typically changes to white. Both conditions reportedly indicate the presence of liver and renal diseases [7, 10]. However, differentiating between Lindsay's and Terry's nails is difficult because these names are used interchangeably in previous research [7, 10]. Furthermore, although aging can affect nail conditions, the prevalence of these changes in the older generation is not well understood [10]. In countries with older populations, nail examination should be considered and investigated for use in clinical situations. Abnormal nail findings should be defined (i.e., a white nail without a lunula) [11]. Currently, there is no information on the distribution of nail changes or whether the changes are transient. Nail changes are more common in the nails of all fingers except the nail of the first finger. Changes can be present mainly in the nail beds of fingers associated with various chronic diseases [11]. Thus, during the physical examination of patients with systemic conditions, physicians can note nail changes on all digits.

Line 68 to 73

White nails can be the result of various biomedical conditions, such as renal and liver disease or a poor nutritional condition, which can be related to morbidity and mortality in patients [11]. Therefore, white nails can be related to patients’ clinical outcomes. However, no studies have investigated the specific relationship between the presence of white nails and patient health outcomes. Nail conditions can change with aging, and the prevalence of white nails can change among the elderly [14].

English language editing is required. 

Response:

Thank you for this comment. Our manuscript has been revised by the English editing company, Editage.

Terry and Lindsay nails are clearly distinguished conditions. 

Response:

Thank you for this remark. We have provided information about the two conditions in the Introduction as follows:

Line 33 to 45

Patients with heart failure, liver cirrhosis, and advanced age experience whitening of the nails, with the distal parts of the nails turning brown, due to telengiectasis; this condition is called Terry's nail [7, 8]. A poor nutritional status can progressively cause Terry’s nail [9]. On the other hand, whitening of the nails with brown distal nail parts due to melanin deposition is called Lindsay's nail; this is indicative of malnutrition and renal disease [10]. The mechanism underlying Terry's and Lindsay’s nails is not fully understood. Given that nail plates and nail bed conditions may not change quickly in response to acute diseases, these changes can be called static physical findings [11]. The state of patients’ nails may be indicative of their nutritional and chronic disease status. Given that nutritional conditions can affect the clinical progression of patients with acute diseases, nail health can be used as a predictive risk factor for patients' morbidity and mortality.

Too many repetitions on white nails in your text. 

Response:

Thank you for this remark. We have revised the text to avoid redundancy.

Reviewer 2 Report

From our point of view, as the authors well point out, the main weakness of the study  is the reliability of the diagnosis of white nails. Recientemente se ha descrito una Application of Transfer Learning using convolutional neural network method for early detection of Terry’s Nail. This method uses Tensorflow Inception-V3 architecture model with the transfer learning method where the results of the experiments that have been done are obtained with 95.24% accuracy.

In future prospective studies that may be carried out by the authors, more precise methods should be used to establish certain diagnoses, thus avoiding the possible bias derived from observation made by trained physicians.

We recommend including the following consideration in the section, as well as the following reference:

doi:10.1088/1742-6596/1201/1/012052

Author Response

Responses to the Reviewers’ comments

Thank you very much for reviewing our manuscript and providing suggestions to improve on our manuscript. We have provided the point-by-point responses to the Reviewers’ comments; our revisions are in red font. Please consider our manuscript for publication.

Reviewer 2

From our point of view, as the authors well point out, the main weakness of the study  is the reliability of the diagnosis of white nails. Recientemente se ha descrito una Application of Transfer Learning using convolutional neural network method for early detection of Terry’s Nail. This method uses Tensorflow Inception-V3 architecture model with the transfer learning method where the results of the experiments that have been done are obtained with 95.24% accuracy.

In future prospective studies that may be carried out by the authors, more precise methods should be used to establish certain diagnoses, thus avoiding the possible bias derived from observation made by trained physicians.

Response:

Thank you for this insightful comment. We have revised the limitations section by adding information about more accurate studies in which transfer learning methods were used.

Line 282 to 288

Another limitation is the reliability of the diagnosis of white nails. In this study, family physicians that were trained to identify white nails performed all diagnoses. We focused only on the disappearance of the lunula and whitening of the proximal part of the nail. As there are variations in the whitening of nails and other types of nails, future studies can investigate the extent of whitening, other nail conditions by using transfer learning methods, and the mortality and morbidity of patients in these contexts [42].

We recommend including the following consideration in the section, as well as the following reference:

doi:10.1088/1742-6596/1201/1/012052

Response:

Line 288

Thank you for this suggestion. We have cited the suggested study where appropriate in the limitations section.